# Reimagining the role of teaching-focused faculty in research-intensive universities: The evolution of scholarly expectations and departmental influence

Alex R. Paine[1]*, Mike Wilton[2], Sabrina M. Solanki[3], Marina Ellefson[4], Julie E. Ferguson[5], Stanley M. Lo[6], Brian K. Sato[3,7]*

1 Department of Biology, College of Arts and Sciences, Baylor University, Waco, Texas, United States of America, 2 Molecular, Cellular, and Developmental Biology, University of California, Santa Barbara, California, United States of America, 3 Division of Teaching Excellence and Innovation, University of California, Irvine, California, United States of America, 4 College of Biological Sciences, University of California, Davis, California, United States of America, 5 Department of Earth System Science, University of California, Irvine, California, United States of America, 6 School of Biological Sciences, University of California, San Diego, California, United States of America, 7 Molecular Biology & Biochemistry, University of California Irvine, Irvine, California, United States of America

☯ These authors contributed equally to this work.
* alex-paine@baylor.edu (ARP); bsato@uci.edu (BKS)

## Abstract

Research-intensive universities aim to conduct cutting-edge research while providing the knowledge and skills necessary to prepare students to excel in their respective fields. As student enrollments surge, many institutions have turned to hiring teaching-focused faculty. In the University of California (UC) system, there exists a unique position known as the Professor of Teaching (PoT). This position is tenure-eligible, and members are required to engage in classroom teaching, scholarly activities, and service responsibilities. To shed light on the background characteristics, roles and perceptions of the impact of teaching-focused faculty in research-intensive institutions, we collected survey data from STEM PoT faculty across the UC system. We employed a mixed methods approach, using descriptive and inferential statistics to analyze quantitative responses and thematic analysis to examine open-ended qualitative data. Our analysis shows that pre-tenure PoTs place greater emphasis on scholarly activities relative to their peers who have been in the role for longer. However, their training and the institutional resources provided may not align with expectations for scholarly activities. Additionally, we find that PoTs who engage in research perceive that they have a more significant impact on their colleagues' teaching. This finding underscores the value of research, even for teaching-focused faculty. This study informs the evolving landscape of teaching-focused faculty within research-intensive universities and provides recommendations for administrators considering how to ensure that their institutions are fulfilling their educational mission.

**Data availability statement:** All relevant data are within the manuscript and its Supporting Information files.

**Funding:** This work was supported by the National Science Foundation Grant DUE #1821724. There was no additional external funding received for this study.

**Competing interests:** The authors have declared that no competing interests exist.

## Introduction

Over the past few decades, there has been a rise in the number of teaching-focused faculty hired within science, technology, engineering, and mathematics (STEM) programs at research-intensive colleges and universities [1]. This has been driven in part in response to calls for increasing the inclusiveness of STEM education in higher education both in the US [2–6] and globally [7–10]. With the widespread interest in improving STEM education, teaching-focused faculty are increasingly being relied on to address issues of instructional quality both through their own teaching as well as by serving as resources for their colleagues and their department's academic programs [11].

Teaching-focused faculty are meant to be specialized both in regard to their professional responsibilities as well as their knowledge and expertise. The most well-established teaching-focused faculty position is that of the adjunct lecturer. As of 2016, nearly 70% of higher education instructors were non-tenure track lectures, making them the predominant form of faculty type across nearly all institutions [12]. In contrast to this more traditional adjunct lecturer position, growing numbers of institutions have also incorporated teaching-focused faculty who have responsibilities that go beyond classroom instruction [13–18]. In addition to instructional duties, these positions may require service and/or scholarly productivity in the educational space, creating faculty that more closely mirror traditional tenure-track research-focused faculty.

The University of California (UC) system's tenure-track, teaching-focused faculty position is the Professor of Teaching (hereafter referred to as PoTs). PoTs comprise roughly 10% of tenure-eligible faculty across the UC system. The promotion criteria for PoT faculty reflect that of the Research Professor but with greater emphasis placed on the value of teaching excellence [19,20]. The PoT role places instructional duties at the forefront, as evidenced by a larger course load and an expected focus on education beyond the classroom (e.g., an increased expectation to participate in pedagogy-focused professional development) [19]. Just as Research Professors are required to engage in scholarship, service, and teaching, PoTs must also pursue each of these endeavors to earn tenure. Like Research Professors, PoTs are eligible for a variety of pre- (Assistant PoT) and post-tenure ranks (Associate PoT and Full PoT).

### Theoretical framework

We consider the roles and potential impacts of PoT faculty in the context of Stoll's [21] *capacity building* framework [22]. This framework is couched within ecological systems theory [23] and describes universities as dynamic and adaptive systems that are influenced by both internal and external forces at a variety of levels. These forces contribute to an institution's internal capacity for educational change, which is defined as the ability of an institution to adapt to the educational needs of their students. The capacity building framework says that internal capacity is influenced by (1) individual faculty within the institution, (2) the institution's social and structural learning context, and (3) the external context (e.g. the broader community or the institution's historical and cultural influences). Each of these influences is then itself informed by a wide variety of factors.

We argue that teaching-focused faculty are uniquely situated to influence an institution's internal capacity to a greater degree than their research-focused colleagues or traditional adjunct lecturers. PoTs have more advanced conceptions of teaching and learning relative to their research-focused counterparts [24], possess self-imposed expectations that they should be engaging in cutting-edge pedagogical activities and professional development opportunities [25], and are more likely to be implementing active learning pedagogies [26]. Within the context of the institution, PoTs play a central role in departmental discussions and influence networks [27]. Additionally, PoTs are increasingly serving in leadership roles. Together, these contexts provide a means by which they can impact student success beyond their individual classrooms.

But there is still much to learn. Many key levers within the capacity building framework remain relatively unknown. These include the extent to which PoTs believe they can make a difference, their sense of interdependence within their departments, their morale, and the history and culture within the institution in regard to the PoT position. For universities to maximize the impact of the PoT position on both student and faculty success, it is essential that we understand the role and the perspectives of PoTs themselves.

### Research on PoT faculty

UC STEM PoT faculty are primarily recruited to ease departmental teaching loads, to ensure consistent and high-quality instruction, and actively contribute to specialized teaching and service activities [18,19]. Over time, administrators have increasingly recognized the broader contributions of these faculty members beyond classroom instruction. This includes their role as pedagogical resources for colleagues, their capacity to secure external funding, and their active participation in education-focused scholarship [18]. As described above, research has identified numerous positive impacts of PoT faculty. These include their use of active learning practices [26], their views of the instructor's role as a facilitator of learning rather than the sole provider of knowledge [24], and their departmental influence in pedagogical discussions related to diversity, equity, and inclusion [27].

Despite these positive impacts, research has also identified significant barriers to faculty in these teaching-focused positions. Administrators noted concerns related to the former PoT position title, *Lecturer with Potential Security of Employment.* They also pointed to a lack of inclusion for PoT faculty, both in terms of their physical office location and their departmental rights and responsibilities related to Research Professors. Together, these factors contributed to a feeling of "second-class citizen" status [18]. It was also noted that the criteria for PoTs to earn tenure were also perceived to be ambiguous, particularly from the perspective of the Research Professors who were primarily tasked with evaluating their success [18]. Research has shown that negative aspects of teaching-focused faculty positions have led to a significant number of these faculty (30–40% from the cited studies) to consider leaving the position, the university, or even higher education altogether [28–30].

In a 2017 survey of UC STEM PoT faculty, it was noted that the roles and responsibilities within this faculty line were not uniformly distributed. Specifically, Assistant PoT respondents indicated allocating a significantly greater portion of their time to scholarly activities compared to their tenured PoT counterparts [19]. PoTs of different rank also had different perceptions of what constituted scholarly activity. Specifically, Assistant and Associate PoTs reported discipline-education research (DBER) to a higher degree relative to those at the Full PoT level [19].

As the popularity of teaching-focused positions continues to increase nationwide [13,28], we re-visited the characterization of the STEM PoT position and the individuals within it to better understand the impact of PoTs on the institution. Specifically, we examined the following research questions through a survey of PoTs across the UC system:

**RQ1:** Do we observe a shift in the time more recently hired PoT faculty spend on scholarly activities and the specific scholarly activities that they choose to engage in?

**RQ2:** How does engagement in scholarly activities correlate with the professional identity and self-reported influence of PoTs?

**RQ3:** How prepared are PoTs to engage in their scholarly responsibilities?

Our mixed methods analysis focuses on positioning PoTs within the context of the research-intensive institution. The mission of these institutions is two-fold: conducting cutting-edge research and providing high quality educational experiences. However, these goals are often not aligned. Prior work has demonstrated that there is little correlation between a faculty member's research excellence and teaching abilities [31–33]. In addition, an institution's finite resources must be divided between research and teaching activities [13,33,34]. Due to the expectation that PoTs engage in both teaching and scholarly work, these faculty have the potential to bridge the university's missions by conducting educational research and then implementing their findings into practice. By better understanding PoTs' perspectives and responsibilities around conducting scholarly activities, we aim to provide concrete recommendations for administrators who have the responsibility of supporting teaching-focused faculty and for junior scholars intending to pursue similar positions.

## Methods

This study was designed with the intent of characterizing perceptions of STEM PoT faculty within the UC system.

### Procedures

**Survey development.** The PoT survey presented here was designed to collect a broad spectrum of data from PoTs, with the goal of gaining a richer understanding of the role. Our analysis centers on five critical areas: demographic data, job responsibilities, resource availability, professional identity, and the impact on colleagues' teaching methods. The methodology employed to construct questions in these domains is elaborated upon below – the majority of which were revised from the previously published iteration of this survey [19]. A preliminary version of the survey was circulated among a dozen PoTs to obtain feedback on the items. This feedback led to revisions of the survey, which were subsequently administered to all UC STEM PoTs as outlined in the sections that follow.

The survey data presented here are part of a broader survey which consisted of 170 questions. For the specific research questions being addressed in this analysis we focused on faculty responses to a subset of questions (24 questions in total), the following areas of the survey will be reported: demographic information, TP/PoT role responsibilities, resources available, perceived influence on colleagues' teaching practices, and professional identity. These questions can be found in the supplemental materials (S1 Table).

**Data collection.** Data was collected from the nine undergraduate-serving campuses in the UC system. Participants were identified through the University of California Office of the President's Academic Personnel office as individuals holding a PoT faculty position. Surveys were distributed to participants via email in the fall of 2021 with information pertaining to the purpose of the survey and its use in this study. The survey was sent using Qualtrics to 473 PoT faculty, the total number of individuals within the position at the time the survey was released. The response rate was 63%. While the survey went out to PoT faculty in all disciplines, we are only including responses from those in STEM fields due to the persistent equity issues that pervade these disciplines, and thus the potential impact that this position can have on addressing these issues. We define STEM according to the National Science Foundation definition, including the disciplines of biological sciences, physical sciences, computer and information sciences, geosciences, engineering, mathematics, and social, behavioral, and economic sciences. While the survey was completed by 298 participants, after removing individuals in non-STEM fields as well as those who did not respond to at least 85% of the survey questions, our sample size was 158 respondents.

**Demographic information.** Participants were asked to provide demographic information including gender identity, ethnicity/race, first generation status, UC campus, department, rank, time in position, and previous training.

As seen in Table 1, 43% of the survey participants identified as cis-gendered females. Most of the survey respondents identified as White (71.5%), with the second largest groups being Asian (5.7%) and Hispanic or Latina/o/x (5.7%). In terms of generational status, 24.1% were first-generation college graduates (defined by individuals whose parents did not complete a four-year degree in the United States). Our data demonstrates that the majority of PoTs are employed across four

**Table 1. Demographic data for STEM Professors of Teaching.**

| | Assistant PoT | | Associate/Full PoT | | Total | |
|---|---|---|---|---|---|---|
| | Count | % | Count | % | Count | % |
| **Faculty Rank** | | | | | | |
| Rank | 77 | 48.7% | 81 | 51.3% | 158 | 100.0% |
| **Gender Identity** | | | | | | |
| Cis-gender male/man | 34 | 21.5% | 41 | 25.9% | 75 | 47.5% |
| Cis-gender female/woman | 35 | 22.2% | 33 | 20.9% | 68 | 43.0% |
| Genderqueer, gender non-binary, transgender, or gender fluid | 1 | 0.6% | 1 | 0.6% | 2 | 1.3% |
| Prefer not to answer | 7 | 4.4% | 6 | 3.8% | 13 | 8.2% |
| **Ethnicity/Race** | | | | | | |
| Asian | 4 | 2.5% | 5 | 3.2% | 9 | 5.7% |
| Black or African American | 1 | 0.6% | 0 | 0.0% | 1 | 0.6% |
| Hispanic or Latina/o/x | 7 | 4.4% | 2 | 1.3% | 9 | 5.7% |
| White | 51 | 32.3% | 62 | 39.2% | 113 | 71.5% |
| Multi-ethnic | 5 | 3.2% | 4 | 2.5% | 9 | 5.7% |
| Other | 2 | 1.3% | 1 | 0.6% | 3 | 1.9% |
| Prefer not to answer | 7 | 4.4% | 7 | 4.4% | 14 | 8.9% |
| **College-Graduate Status** | | | | | | |
| First-Generation College Graduate | 14 | 8.9% | 24 | 15.2% | 38 | 24.1% |
| **University of California Campus** | | | | | | |
| Campus 1 | 18 | 11.4% | 17 | 10.8% | 35 | 22.2% |
| Campus 2 | 11 | 7.0% | 20 | 12.7% | 31 | 19.6% |
| Campus 3 | 10 | 6.3% | 14 | 8.9% | 24 | 15.2% |
| Campus 4 | 13 | 8.2% | 9 | 5.7% | 22 | 13.9% |
| Campus 5 | 6 | 3.8% | 6 | 3.8% | 12 | 7.6% |
| Campus 6 | 7 | 4.4% | 3 | 1.9% | 10 | 6.3% |
| Campus 7 | 6 | 3.8% | 3 | 1.9% | 9 | 5.7% |
| Campus 8 | 1 | 0.6% | 5 | 3.2% | 6 | 3.8% |
| Campus 9 | 4 | 2.5% | 3 | 1.9% | 7 | 4.4% |
| **Discipline (Home Department)** | | | | | | |
| Biological Sciences | 25 | 15.8% | 23 | 14.6% | 48 | 30.4% |
| Computer Science/Engineering | 20 | 12.7% | 27 | 17.1% | 47 | 29.7% |
| Social Sciences | 12 | 7.6% | 5 | 3.2% | 17 | 10.8% |
| Physical Sciences | 13 | 8.2% | 20 | 12.7% | 33 | 20.9% |
| Other STEM | 6 | 3.8% | 5 | 3.2% | 11 | 7.0% |

Demographic information for all survey participants (N = 158). For college graduate status, the options were first generation college graduate or non-first-generation college graduate. For discipline, "other STEM" encompasses pharmaceutical sciences and statistics.

of the nine undergraduate-serving UC campuses, with these accounting for 70.9% of employed PoTs in our sample. PoTs are distributed across STEM disciplines, with most respondents being concentrated in the biological sciences (30.4%) computer sciences/engineering (29.7%), physical sciences (20.9%), and social sciences (10.8%); and the remaining 7.0% in other STEM disciplines (which include pharmaceutical sciences and statistics). The majority of the faculty in this position have been hired since 2015 (S1 Fig).

**PoT responsibilities.** These items intended to understand participants' perception of the PoT job expectations. Participants were asked to provide an approximation of the percentage of time they believed they were expected to

participate in the following professional domains: teaching, scholarship, and service, as well as the percentage of time they actually were spending. Additionally, we acquired information on the types of scholarly activities they engaged in (e.g., generating peer reviewed publications, developing undergraduate curriculum, providing professional development for graduate students). Respondents could also provide information about whether or not they perceived any change in expectations to engage in these activities in an open-ended response section.

**Professional identity.** Participants were asked about the degree to which they identified as an instructor and separately as a researcher. This was reflected through a 7-point Likert-scale measure of the degree to which one's professional identity overlapped with that of an instructor or researcher with one representing no overlap of personal identity and identity as an instructor or researcher and 7 being complete overlap with that of an instructor or researcher.

**Influence on colleagues' teaching.** To determine to what degree PoTs believed they influenced their colleagues, we included three Likert-scale items to measure their perceived influence on colleagues' teaching beliefs, knowledge, and practices.

## Training and resources

This portion of the survey was intended to gather information on the types of resources available to those within the PoT position such as training and support around scholarly activities. Information was gathered through a selection menu, allowing survey participants to identify all applicable resources. Additionally, an open-ended response section was included to capture additional insights regarding these resources. To collect data about the formal training of PoTs, survey participants were prompted to specify the kinds of training they have undergone, both within their specific STEM fields and in education research fields. The questionnaire allowed respondents to select all applicable types of training. Formal discipline training within STEM fields included earning a graduate degree (Ph.D., master's degree) or working as a postdoctoral scholar. Formal education research training included earning a graduate degree or postdoctoral experience in either education or discipline-based education research.

## Data analysis

This study employed a mixed methods approach, integrating exploratory quantitative analyses with phenomenologically informed qualitative analysis to investigate faculty experiences across career stages. The qualitative component focused on capturing participants' lived experiences and perceptions, while the quantitative analyses provided a broader contextual understanding of patterns across faculty subgroups.

All quantitative analyses, including two-sample t-tests and multiple regression analyses, were conducted in R [35]. To compare the responses for a variety of survey items from newer faculty (Assistant PoT) to tenured faculty (Associate or Full PoT), a two-sample t-test was used to assess any statistically significant differences between the two groups. The decision to use two-sample t-tests reflects the exploratory nature of the analysis, where the goal is to examine potential differences between new hires (Assistant PoTs) and tenured faculty (Associate/Full PoTs) without specifying the direction of the expected differences beforehand. This approach provides a more open-ended examination of differences between new hires and tenured faculty, contributing to a richer understanding of the factors influencing faculty experiences in various stages of their careers.

Multiple regression analyses were run to explore the relationship between an individuals' research identity and engaging in scholarly activities as well as the relationship between an individuals' influence on colleagues teaching and engaging in scholarly activities. Predictor variables included engaging in scholarly activities, faculty's reported time spent on scholarly activities, generation of peer-reviewed publications, engagement in DBER, mentorship of undergraduate/ graduate student researchers (all were used as dichotomous indicators of whether faculty participated in each of the listed scholarly activities), gender (dummy coded with male as the reference group), race/ethnicity (dummy coded white as the reference group), campus (dummy coded with Campus 1 as the reference group), department (dummy coded with

biological sciences as the reference group), and faculty rank (a dichotomous variable with Assistant PoTs as the reference group).

The full models, which included both scholarly activities and demographic/contextual controls, are presented in the Supplement (S2 & S4 Tables). To evaluate the robustness and parsimony of the findings, reduced models including only the scholarly activity predictors are also reported in the Supplement (S3 & S5 Tables). These reduced models help clarify the unique contributions of scholarly activities without adjustment for background characteristics.

Qualitative coding was used to analyze open-ended responses for questions focusing on availability of resources, opportunities for research skill development, and perceived changes to the PoT role. Through an iterative coding process, emergent codes were identified and categorized to determine frequency of response types.

### Ethics statement

Informed consent was obtained from all individual participants included in the study. Written consent was obtained electronically through a consent form that participants had to read and agree to before proceeding with the survey. No participants under the age of 18 were included in the study. All personal information was removed from the collected data prior to analysis to ensure participants were fully deidentified and their anonymity was preserved.

All data were collected in accordance with the University of California Irvine's Institutional Review Board (UCI IRB Protocol #1976) during the Fall 2021 quarter (August-October 2021).

## Results

### RQ1: Do we observe a shift in the time more recently hired PoT faculty spend on scholarly activities and the specific scholarly activities that they choose to engage in?

We began by examining the percentage of time PoTs perceived they were *expected to spend* on teaching, scholarly activities, and service – which on average was 63%, 19%, and 18% respectively (Table 2A). We then compared responses by pre-tenure and tenured respondents to note if there has been a significant change in recent years to the expectation of this role. For this analysis we used non-tenured faculty as a proxy for recent change. While there was not a difference between the expected time spent on service and teaching between Assistant versus Associate/Full PoTs, Assistant PoTs reported that they are expected to spend significantly more time on their scholarly activities, aligning with prior results (two-sample t-test, $t(165.5) = 2.77$, p = 0.006).

Table 2. Professors of Teaching time spent on scholarly activities, service, and teaching.

| A. | Assistant PoT | Associate/Full PoT | | |
|---|---|---|---|---|
| | % Time | % Time | t(df) | p-value |
| Scholarship | 19.76 (±8.18) ** | 16.08 (±8.97) | 2.77 (165.5) | 0.006** |
| Service | 15.14 (±7.58) | 17.02 (±7.78) | −1.58 (163.7) | 0.116 |
| Teaching | 65.10 (±12.34) | 66.90 (±11.22) | −0.98 (157.1) | 0.328 |
| B. | Assistant PoT | Associate/Full PoT | | |
| | % Time | % Time | t(df) | p-value |
| Scholarship | 18.08 (±11.11) * | 14.59 (±11.07) | 2.03 (162.5) | 0.044* |
| Service | 16.28 (±9.41) | 22.91 (±11.78) *** | −4.05 (165.0) | 7.852E-05*** |
| Teaching | 65.64 (±14.15) | 62.50 (±14.99) | 1.40 (164.8) | 0.165 |

(A) Expected time and (B) actual time spent on scholarly activities, service, and teaching.

These values are reported as average percentages (combined out of 100%). The standard deviation is presented in parenthesis. Two-sample t-test are used to report mean differences between new hires (Assistant TP/PoT) and tenured (Associate/Full TP/PoT) faculty (*p < .05, **p < .01, ***p < .001). An α level of .05 was used for all comparisons. Because six planned comparisons were conducted, we applied a Holm correction for multiple testing.

We followed up this question by identifying the percentage of time that respondents in the various ranks *actually spend* on their teaching, scholarly activities, and service. Assistant PoTs reported that they spend significantly more time on scholarly activities (two-sample t-test, *t(162.5) = 2.03,* p = 0.044) than their tenured colleagues; however, this difference did not remain significant after correction of multiple comparisons (Table 2B). By contrast, Associate/Full PoTs reported spending significantly more time on service (two-sample t-test, *t(165.0) = −4.05,* p < 0.001). Teaching time did not differ significantly between groups.

These findings were reinforced through responses to the following open-ended question found within the survey, "How has the reality of being a PoT been different from your expectations from when you were hired?" Representative responses included:

"The research expectations have been much more than I expected." – *Assistant PoT*

"The expectations in the [Academic Personnel Manual] have changed since I was hired to include a larger role for research/scholarly activity." – *Full PoT*

"Expectations for this position have changed a great deal over the years - now an increased expectation for scholarly activity and service. But teaching is still considered the number one priority" – *Full PoT*

These responses highlight an increase in expectations around scholarly activities, specifically conducting research, since the respondents were hired. While mentioned by both pre-tenure and tenured PoTs, almost twice as many Associate/Full PoTs noted this increased expectation for scholarly activities in their open-ended response.

As the percentage of time spent on scholarly activities seemed to be a key difference between pre-tenure and tenured PoTs, we were curious as to the specific activities these individuals engaged in (Table 3). At α = .05, Assistant PoTs were more likely to report conducting education or discipline-based education research (t(166.5) = 3.67, p < 0.001). In contrast, Associate/Full PoTs were more likely to report providing professional development for graduate students (t(164.4) = −2.19, p = 0.03) and K-12 teachers (t(150.9) = −2.83, p = 0.005), as well as developing undergraduate curriculum (t(143.7) = −2.55, p = 0.01). However, after correction for multiple comparisons, only DBER and k-12 professional development remained significant. PoTs across all ranks reported similarly engaging in discipline-specific research, generating publications, and mentoring undergraduate and graduate student research.

**Table 3. Respondents' self-reported scholarly activities.**

| Accomplished Activities | Assistant PoT | Associate/ Full PoT | t(df) | p-value |
|---|---|---|---|---|
| Discipline-specific research | 46.2% | 41.6% | 0.71 (162.30 | 0.476 |
| Discipline-based education research or education research | 79.5% | 53.9% | 3.67 (166.5) | 0.0003*** |
| Mentoring undergraduate/graduate student research | 70.5% | 77.5% | −0.94 (158.3) | 0.350 |
| Generating peer-reviewed publications | 59.5% | 62.9% | 1.47(167.4) | 0.143 |
| Improving teaching practices in the department/application of evidence-based teaching practices | 69.2% | 79.8% | −1.46 (154.8) | 0.147 |
| Assessment of teaching/education in the department/campus | 44.9% | 62.9% | −2.33 (161.6) | 0.02* |
| Providing professional development for graduate students | 37.2% | 52.8% | −2.19 (164.4) | 0.03* |
| Providing professional development for K–12 teachers | 6.4% | 20.2% | −2.83 (150.9) | 0.005** |
| Developing undergraduate curriculum | 67.9% | 85.4% | −2.55 (143.7) | 0.011* |

PoTs' self-reported whether they did or did not engage in the above activities. The percentage of survey respondents in each category is reported. Two-sample t-test are used to report mean differences between new hires (Assistant PoT) and tenured (Associate/Full PoT) faculty (*p < 0.05, **p < 0.01, ***p < 0.001). An α level of 0.05 was used. Because nine planned comparisons were conducted, we applied a Holm correction for multiple testing.

**Research question #2: How does engagement in scholarly activities correlate with the professional identity and self-reported influence of PoTs?**

**Professional identity.** Thus far, differences have been identified in perceived behaviors amongst non-tenured and tenured teaching faculty, particularly in relation to the scholarly activity component of the position. To better understand how these differences may be related to perceptions of self, respondents were asked to report the degree to which they identified as an instructor and a researcher.

Assistant and Associate/Full PoTs reported nearly identical perceptions when it comes to their identity as instructors (Table 4). This is in alignment with the PoT position being primarily teaching-focused. However, when asked about their identity as researchers, Assistant PoTs identify significantly more as researchers than their more senior colleagues (p = 0.006).

To better understand the difference in research identity for Assistant and Associate/Full PoTs, a multiple regression analysis was run to uncover factors contributing to research identity (S2 Table). Engaging in DBER was a significant predictor of research identity ($\beta$ = 0.88, p = 0.01), but engagement in other scholarly activities did not predict their research identity, nor did an individual's instructor identity, demographics, or their UC campus. Although demographic variables did not significantly contribute to the model, they were retained to control for potential background influences. A reduced model including only the scholarly activity predictors produced consistent results and is also presented in the supplemental materials (S3 Table). A Type III ANOVA confirmed that DBER contributed significant unique variance to the model ($F(1,122) = 7.57$, $p = 0.007$), whereas no other predictors reached statistical significance.

**Influence on teaching.** To capture overall teaching influence, we used an average composite score based on faculty self-reports of their influence on colleagues' teaching beliefs, knowledge, and practices. This model (S4 Table) revealed significant positive associations between overall teaching influence and engaging in DBER ($\beta$ = 0.44, p = 0.03) and generating peer-reviewed publications ($\beta$ = 0.36, p = 0.01). A significant negative association was observed with mentoring undergraduate or graduate student researchers ($\beta$ = −0.45, p = 0.02). A reduced model including only the scholarly activity predictors produced consistent results and is presented in the supplemental materials (S5 Table).

To further support the interpretation of these models, we conducted Type III ANOVAs to evaluate the overall model structure and the unique contribution of each predictor. The ANOVA for this model revealed significant effects of DBER (DBER) ($F(1,135) = 6.54$, $p = 0.012$), mentoring undergraduate or graduate student researchers ($F(1,135) = 6.62$, $p = 0.011$), and generating peer-reviewed publications ($F(1,135) = 10.07$, $p = 0.002$) on increased perceived influence on colleagues' teaching.

**RQ3: How prepared are PoTs to engage in their scholarly responsibilities?**

**Training.** As noted above, the majority of Assistant PoT faculty reported conducting discipline-based education research (79.5% compared to 59.8% of PoTs overall). It is therefore important to understand whether PoTs are coming

**Table 4. Teaching Professors' identity as instructors and researchers.**

| | Average Alignment Score | |
|---|---|---|
| **Identity alignment with:** | **Assistant PoT** | **Associate/Full PoT** |
| Being an instructor | 5.84 (± 1.12) | 5.83 (± 0.99) |
| Being a researcher | 4.00 (± 1.65) ** | 3.28 (± 1.59) |

A value of 1 indicated no overlap between an individual's identity and that of a researcher while a 7 indicated complete overlap. These values reported are the mean responses for the given group. The standard deviation is presented in parenthesis. Two-sample t-test are used to report mean differences between non-tenured (Assistant PoT) and tenured (Associate/Full PoT) faculty (*p < 0.05, **p < 0.01, ***p < 0.001).

into this role with relevant training. Examination of the background training of PoT faculty revealed the majority earned a PhD and postdoctoral training in their departmental STEM discipline. In contrast, only about 10% of PoTs reported earning an undergraduate or graduate degree in an education research field and 5% with postdoctoral experience (Table 5). These trends are consistent across PoTs of varying ranks, although Assistant PoTs are slightly more likely to have post-doctoral experience in an education research field (10.4% of Assistant PoT faculty as compared to 2.8% of their tenured colleagues; p = 0.07).

**Resources.** As the overwhelming majority of PoTs do not have formal training in DBER, it is important to understand whether these individuals believe the university provided them with the necessary resources for success in this, and other, scholarly endeavors.

On average, PoTs reported receiving $38,700 (±$30,322) in start-up funds. Assistant PoTs received significantly more (two-sample t-test, p < .01) start-up funds upon hire – an average of $45,400 versus the average $29,000 that their ten-ured colleagues received. When surveyed about lab space, 51.2% of respondents reported needing lab space (55% of Assistant PoTs versus 47.6% of tenured faculty). Of those who reported needing lab space, only 37% of these individuals received it (34% of Assistant PoTs versus 42.5% of tenured faculty).

We then asked PoTs about other types of support available to them to pursue scholarly work (Table 6). Many reported being able to supervise undergraduates (80.7%), having access to sabbatical (77.8%), having access to materials support in the form of equipment (45.5%), and receiving financial support to attend conferences/workshops (45.5%). Less common forms of support were access to postdoctoral scholars (20.5%) and scheduled reductions in their teaching responsibilities (16.8%). No significant difference was observed between the different faculty ranks.

While many respondents noted that these resources were available, the open-ended responses told a more complicated story:

"All of these [referring to the dropdown options of resources] are available to me, though the feasibility of obtaining them are more or less challenging (e.g., I'd need a large grant to be able to pay a postdoc... but I am technically "able" to)."

"All of these are possible, but in practice these are limited resources."

"Sabbatical is technically available but will be extremely difficult to take advantage of in practice."

These comments highlight that the technical availability of these resources did not necessarily translate to them being available in practice. Furthermore, when prompted to report on the availability of campus opportunities to improve their skills related to scholarly work, 60% of PoTs reported that these resources were inadequate (no significant difference

**Table 5. Teaching Professors formal discipline-specific and educational training.**

| | Assistant PT/PoT | | Associate/Full TP/PoT | |
| --- | --- | --- | --- | --- |
| | % | Count | % | Count |
| **Discipline Training** | | | | |
| Post Doc | 38.2% | 29 | 47.5% | 38 |
| PhD | 96.1% | 73 | 93.8% | 75 |
| Master's | 1.3% | 1 | 3.8% | 3 |
| **Education/Education Research Training** | | | | |
| Education Post Doc | 10.4%[+] | 7 | 2.8% | 2 |
| Education PhD | 10.4% | 7 | 6.9% | 5 |
| Education Master's | 2.8% | 1 | 9.7%* | 7 |

PoTs' self-reported formal discipline-specific and education/education research training including postdoctoral training, Ph.D., or master's degree. The percentage and count of survey respondents in each category is reported. Two-sample t-test are used to report mean differences between new hires (Assistant PoT) and tenured (Associate/Full PoT) faculty (+p = 0.07, *p < 0.05).

**Table 6. Perception of resources and sources of support available to pursue scholarly work.**

| | Assistant PoT | | Associate/Full PoT | |
| --- | --- | --- | --- | --- |
| | % | Count | % | Count |
| Undergraduates working under your supervision | 80.5% | 62 | 82.1% | 69 |
| MS Graduate students working under your supervision | 39.0% | 30 | 41.7% | 35 |
| PhD graduate students working under your supervision | 33.8% | 26 | 32.1% | 27 |
| Postdoctoral scholars working under your supervision | 19.5% | 15 | 17.9% | 15 |
| Materials support in the form of equipment | 45.5% | 35 | 44.0% | 37 |
| Financial support to attend conferences and/or workshops | 45.5% | 35 | 50.0% | 42 |
| Sabbatical | 67.5% | 52 | 81.0% | 68 |
| Scheduled reduced teaching responsibilities | 11.7% | 9 | 19.0% | 16 |
| Other | 7.8% | 6 | 9.5% | 8 |

PoTs were asked to report whether or not the above resources were available for the pursuit of scholarly work. The percentage and count of survey respondents in each category are reported. Two-sample t-tests are used to report mean differences between non-tenured (Assistant PoT) and tenured (Associate/Full PoT) faculty.

between non-tenured and tenured faculty responses). In comparison, only 7.5% of PoTs reported that campus opportunities to improve their teaching were inadequate.

## Discussion

There has been considerable literature, including several national reports from influential organizations [4,7,9,36–38], that have highlighted the need to improve student outcomes in higher education STEM programs. In response, research has identified a variety of potential interventions to create more inclusive and equitable learning environments at the student, course, and institution level. One way to aid the implementation of these interventions could be through the integration of teaching-focused faculty into STEM departments [37,39,40]. Through the lens of capacity building framework [21,22], we speculate that these faculty may be particularly impactful. We posit that these faculty could strengthen an institution's internal capacity for educational change. Many of the factors in this framework that influence institutional capacity are particularly relevant to teaching-focused faculty. For example, respondents described engaging in, or being expected to engage in, a range of education-focused scholarly activities (Table 3). Such activities align with the "motivation to learn" factor within the capacity building framework. Additionally, we captured varying degrees of self-reported influence, which indicates a "sense of interdependence" for these faculty, another factor within the framework.

As teaching-focused faculty are becoming more prominent in higher education, it is essential that we better understand how to leverage and support these individuals. While our results are presented in the context of a position specific to the University of California, the conclusions and recommendations from this study can be implemented more broadly.

### The PoT role begins to evolve with new expectations

To contextualize our survey results, we first consider a prior study on PoTs [19] that characterized faculty in this role. That study revealed nuanced shifts in the position over time [19], offering a valuable snapshot that laid the groundwork for future investigations into the potential institutional impacts of these individuals. Since the initial survey, the number of faculty appointments in the PoT role has notably increased (S1 Fig); however, the demographic composition of these faculty members has remained relatively consistent. There has been a small, but significant, rise in education research postdoctoral training among the Assistant PoTs. As seen by Harlow *et al.* [19], we also found a notable difference in the percentage of time Assistant PoTs believe faculty in the PoT position are expected to allocate and actually allocate to scholarly activities. Assistant PoTs are also more likely to engage in discipline-based education research as a form of scholarship

relative to their tenured peers, mirroring the findings of the previous survey [19]. This may reflect both an increased interest among these faculty to engage in DBER as well as changing expectations of stakeholders responsible for their hire or individuals involved in evaluating PoTs for merit and promotion purposes. On the other hand, Associate/Full PoTs invest significantly more time in service activities. The increase in the service commitments observed in tenured faculty is likely attributable to the expectation that service responsibilities expand in tandem with faculty rank and seniority [40–42].

### Influence of scholarly activities on teaching-focused faculty's identity

This study sheds light on the unique professional identity dynamics among teaching-focused faculty at research-intensive institutions, introducing a nuanced perspective on the interplay between teaching and research identities. The study reveals that Assistant PoTs identify significantly more as researchers than their tenured colleagues, even though both groups reported similar degree of instructor-identity. This challenges the previously described notion that a strong research identity conflicts with a teaching identity [43,44]. This perception of a single dominant identity is also present in our context, as stakeholders are often surprised by PoT involvement in research. Despite this, stakeholders continue to view teaching-focused faculty as instructors whose function is overwhelmingly to reduce the departmental teaching load [18]. We speculate that the dual teaching/research identity may be specific both to the PoT position as well as the institutional context in which this position is found (research-intensive university). It would be interesting to examine how these identities have changed over time, as they likely reflect a shift in the perspectives and values of the university. A university's history is another factor within the capacity building framework [21,22], and understanding the broader causes for this research emphasis may help administrators consider future roles for teaching-focused faculty. Future work can also examine identity formation more broadly in STEM to see how experiences and institutional structures influence how graduate students, postdoctoral scholars, and faculty perceive their teaching and research identities.

### Teaching-focused faculty's scholarship potentially positions them to influence pedagogy and instruction

Teaching-focused faculty have the potential to address issues in STEM education through innovation in their own teaching as well as by serving as pedagogical resources [11]. In this context, we found a positive correlation between faculty engagement in scholarly activities (specifically generating peer-reviewed publications and participation in DBER) and their self-reported influence on their colleagues' teaching. This finding aligns with prior research that highlights the benefits of engaging in DBER. Such work provides instructors with the information they need to make informed decisions about pedagogy, instructional materials, and assessment practices, leading to more relevant and effective education [45].

We also observed a negative correlation between a faculty member's involvement in mentoring undergraduate and graduate student research and their impact on colleagues' teaching. This correlation may be attributed to the presence of competing demands and priorities. Faculty members who are deeply engaged in student research may invest substantial time and energy in this pursuit. Additionally, the nature of the mentee's research can further compound these demands. If faculty members spend a significant amount of time and energy on mentoring discipline-specific research, it may diminish their capacity to participate in activities like DBER. As a result, their efforts may not directly align with influencing colleagues' teaching. These relationships between faculty are key influencers of internal capacity. Identifying ways to build positive collaborations can have a disproportionate impact on an educational program relative to the influence of a single faculty member. Teaching-faculty influence may also signal a broader cultural shift. These individual interactions may in fact be a sign that the department or university as a whole is more open to innovation in the educational space.

### Recommendations for hiring

An individual's life and career experiences shape their priorities as a faculty member, which in turn can influence an institution's internal capacity. A more recent emphasis on research *expectations* within the PoT position highlights a

misalignment with PoT hiring, as the overwhelming majority of those in the position do not have formal training in this arena. While there was a slight increase in Assistant PoTs with an education or discipline-based education research PhD relative to Associate/Full PoTs (10.4% vs. 2.8%), this is clearly not a consistent requirement for those being hired in the role. Although a lack of education research training does not disqualify PoTs from pursuing such research, it does put them at a disadvantage both in terms of their abilities to publish peer-reviewed work as well as obtain extramural funding. As such, we recommend hiring committees consider the importance of education research training if the PoT being hired is expected to engage in this practice.

For junior scholars interested in pursuing a teaching-focused faculty position, there appears to be value in having formal education-research training during one's graduate or postdoctoral experience. This aligns with the increased frequency of education research being integrated into more traditional discipline-focused research programs [5,45,46].

Similarly, to conduct research, the proper resources must be provided. Beyond increased professional development opportunities, PoTs who engage in research reported that they need more time. This need could be provided through teaching relief and physical space. This last point is purposefully underscored as there is a misconception that education research does not require additional time or space. The assumption is that it can be conducted within the classroom [1]. This ignores the fact that many education research studies are not classroom-related, and that a thriving research program requires a team of individuals who have the space to conduct their work in a collaborative fashion.

## Limitations

While this survey was sent to all individuals within the PoT position across the University of California, a self-selected group chose to complete the survey. While this group was representative of the distribution of PoT faculty by UC campus and discipline, we cannot claim that the survey responses fully represent the entirety of the PoT population. Additionally, self-reported studies where individuals provide information about themselves, their behaviors, their perceived expectations, and their perception of influence, have several limitations that can impact the reliability and validity of the data collected. For example, in this specific study, there might be a tendency for participants to feel compelled to align their reported accomplishments with their perceived expectations. As influence was self-reported, having additional measures, including surveying departmental colleagues and measuring more objective outcomes like teaching practices, would add additional data to complement our findings. Related to this, our findings regarding the relationship between research activity and influence are only correlational and potentially reflect characteristics of a PoT who conducts research or identifies as a researcher and is not related to the research activity itself.

## Conclusion

Research-intensive institutions are invested both in conducting innovative research and developing the next generation of future scientists and leaders. Unfortunately, these two aims are often not complementary. This distinction in many cases is amplified by the merit or promotion process where faculty success is measured primarily by research excellence. However, by encouraging the research efforts of teaching-focused faculty, we may be able to bridge the gap between the research-focused and teaching-focused aims of the university, and in doing so, increase the influence teaching-focused faculty have on their colleagues' teaching. As this form of influence has been identified as one of the more significant goals of hiring PoT faculty, we believe that fostering an environment that promotes faculty success in their scholarly activities may also promote STEM student success.

## Supporting information

**S1 Table. Survey Questions and Response Types.**
(DOCX)

**S2 Table. Scholarly activities as predictors of an individuals' research identity.**
(DOCX)

**S3 Table. Reduced Regression model for Research Identity.** Reduced regression model including only scholarly activity predictors of research identity. This model is presented for comparison with the full model (S2 Table). Demographic and contextual controls were excluded in this model.
(DOCX)

**S4 Table. Scholarly activities as predictors of an individuals' perceived influence on teaching.**
(DOCX)

**S5 Table. Reduced Regression model for Perceived Influence on Teaching.** Reduced regression model including only scholarly activity predictors of perceived influence on teaching. This model is presented for comparison with the full model (S3 Table). Demographic and contextual controls were excluded in this model.
(DOCX)

## Author contributions

**Conceptualization:** Mike Wilton, Julie E. Ferguson, Brian K. Sato.

**Formal analysis:** Alex Renee Paine.

**Writing – original draft:** Alex Renee Paine.

**Writing – review & editing:** Mike Wilton, Sabrina M. Solanki, Marina Ellefson, Julie E. Ferguson, Stanley M. Lo, Brian K. Sato.

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
