## [Decision Letter · Decision Letter 0]

21 Apr 2025

Dear Dr. Paine,

Thank you for submitting your manuscript to PLOS ONE. After careful consideration, we feel that it has merit but does not fully meet PLOS ONE’s publication criteria as it currently stands. Therefore, we invite you to submit a revised version of the manuscript that addresses the points raised during the review process.

**Thanks for submitting you article to PLOS ONE. It has been reviewed by two individuals, one who has a review similar to mine and one accept. You need to check the Information for Authors for citations, formatting etc as your manuscript  does not follow the guidelines.**
**In addition, it is difficult to understand your analyses as presented. Finally, in the discussion and limitations from the findings it seems to me that these teaching professors are not really teaching professors. --they just needed a job. They are more similar to research professors without the support. They are more like those who select to go to traditional teaching colleges.  Please discuss the possibility  of this in your discussion.**

We look forward to receiving your revised manuscript.

Kind regards,

Mary Diane Clark, PhD

Academic Editor

PLOS ONE

**Journal Requirements:**

Please ensure that your manuscript meets PLOS ONE's style requirements, including those for file naming. The PLOS ONE style templates can be found at https://journals.plos.org/plosone/s/file?id=wjVg/PLOSOne_formatting_sample_main_body.pdf and https://journals.plos.org/plosone/s/file?id=ba62/PLOSOne_formatting_sample_title_authors_affiliations.pdf 2. Thank you for stating in your Funding Statement: This work was supported by the National Science Foundation Grant DUE #1821724.  Please provide an amended statement that declares *all* the funding or sources of support (whether external or internal to your organization) received during this study, as detailed online in our guide for authors at http://journals.plos.org/plosone/s/submit-now.  Please also include the statement “There was no additional external funding received for this study.” in your updated Funding Statement. Please include your amended Funding Statement within your cover letter. We will change the online submission form on your behalf. 3. Thank you for stating the following in the Acknowledgments Section of your manuscript: This work was supported by the National Science Foundation Grant DUE #1821724. We note that you have provided funding information that is not currently declared in your Funding Statement. However, funding information should not appear in the Acknowledgments section or other areas of your manuscript. We will only publish funding information present in the Funding Statement section of the online submission form. Please remove any funding-related text from the manuscript and let us know how you would like to update your Funding Statement. Currently, your Funding Statement reads as follows: This work was supported by the National Science Foundation Grant DUE #1821724.   Please include your amended statements within your cover letter; we will change the online submission form on your behalf. 4. We note that this data set consists of interview transcripts. Can you please confirm that all participants gave consent for interview transcript to be published? If they DID provide consent for these transcripts to be published, please also confirm that the transcripts do not contain any potentially identifying information (or let us know if the participants consented to having their personal details published and made publicly available). We consider the following details to be identifying information:- Names, nicknames, and initials- Age more specific than round numbers- GPS coordinates, physical addresses, IP addresses, email addresses- Information in small sample sizes (e.g. 40 students from X class in X year at X university)- Specific dates (e.g. visit dates, interview dates)- ID numbers Or, if the participants DID NOT provide consent for these transcripts to be published:- Provide a de-identified version of the data or excerpts of interview responses- Provide information regarding how these transcripts can be accessed by researchers who meet the criteria for access to confidential data, including:a) the grounds for restrictionb) the name of the ethics committee, Institutional Review Board, or third-party organization that is imposing sharing restrictions on the datac) a non-author, institutional point of contact that is able to field data access queries, in the interest of maintaining long-term data accessibility.d) Any relevant data set names, URLs, DOIs, etc. that an independent researcher would need in order to request your minimal data set. For further information on sharing data that contains sensitive participant information, please see: https://journals.plos.org/plosone/s/data-availability#loc-human-research-participant-data-and-other-sensitive-data If there are ethical, legal, or third-party restrictions upon your dataset, you must provide all of the following details (https://journals.plos.org/plosone/s/data-availability#loc-acceptable-data-access-restrictions):a) A complete description of the datasetb) The nature of the restrictions upon the data (ethical, legal, or owned by a third party) and the reasoning behind themc) The full name of the body imposing the restrictions upon your dataset (ethics committee, institution, data access committee, etc)d) If the data are owned by a third party, confirmation of whether the authors received any special privileges in accessing the data that other researchers would not havee) Direct, non-author contact information (preferably email) for the body imposing the restrictions upon the data, to which data access requests can be sent 5. Please include your full ethics statement in the ‘Methods’ section of your manuscript file. In your statement, please include the full name of the IRB or ethics committee who approved or waived your study, as well as whether or not you obtained informed written or verbal consent. If consent was waived for your study, please include this information in your statement as well. 6. Please include captions for your Supporting Information files at the end of your manuscript, and update any in-text citations to match accordingly. Please see our Supporting Information guidelines for more information: http://journals.plos.org/plosone/s/supporting-information.

**Additional Editor Comments:**

Important topic as many research trained PhDs are accepting jobs as Teaching faculty most with as little educational backgrounds as those in tenure track research positions.

Reviewer 1 and I have provided many suggestions for improving the manuscript. I have some concerns about how the analyses were conducted as what is provided in the paper is somewhat limited to allow me to evaluate how the statistics were conducted. To avoid this you Data Analyses section needs to be much more detailed to allow the reader to understand your analysis

Please read the comments in the attached file.

In addition, there are three sections that I suggest you eliminate. Figure 1 takes a lot of space and could be conveyed easier in text or at most a table. DEIi is not the focus on this manuscript, I suggest you take it out. Then Brofenbrenner does not contribute to this manuscript.

Reviewers' comments:

Reviewer's Responses to Questions

**Comments to the Author**

1. Is the manuscript technically sound, and do the data support the conclusions?

Reviewer #1: Partly

Reviewer #2: Yes

2. Has the statistical analysis been performed appropriately and rigorously?

Reviewer #1: I Don't Know

Reviewer #2: Yes

3. Have the authors made all data underlying the findings in their manuscript fully available?

Reviewer #1: Yes

Reviewer #2: Yes

4. Is the manuscript presented in an intelligible fashion and written in standard English?

Reviewer #1: Yes

Reviewer #2: Yes

**Reviewer #1: ** The manuscript contains minor grammatical errors that require correction for improved clarity and readability.

There are inconsistencies in the formatting and usage of in-text citations that should be addressed.

The abstract does not include mention of the qualitative or statistical methods used in the study, which may limit readers' understanding of the research design.

Up to the Methods section, the manuscript lacks any discussion of statistical analysis, which makes the later emergence of a mixed methods approach feel abrupt and disjointed. Earlier sections read as though the study was exclusively qualitative.

The Discussion section opens with a focus on diversity, equity, and inclusion (DEI), but these themes are not introduced or addressed elsewhere in the manuscript. As a result, this section appears disconnected and underdeveloped.

In the Data Analysis section, the investigators do not address potential statistical relationships between participants’ gender identity and their perceptions of their roles as STEM Teaching Faculty members. Given the near-even gender distribution of participants and the male-dominated nature of STEM fields, exploring this relationship may yield valuable insights.

The research questions are not accompanied by explicitly stated hypotheses. In a mixed methods investigation, articulating at least tentative expectations would strengthen the study’s rationale and coherence.

For RQ3, the discussion of participants’ influence on colleagues’ teaching lacks cohesion and depth. Additionally, the basis for measuring this influence is unclear, as no data were collected from the colleagues themselves, only three Likert-scale questions of the participants' opinions on their influence were presented. This section appears hihgly speculative, potentially misleading, and would benefit from further investigation, elaboration and clarification.

**Reviewer #2:**  This article highlights the increased reliance of some universities, in this case, within the UC system in STEM fields, upon teaching faculty, that these faculty are expected to and are engaging in significant pedagogy research, and that their pedagogically informed methods are perceived to impact teaching improvements and departmental teaching strategies.

The article clearly elaborates theoretical framework, methods, and data collection, as well as results demonstrating that teaching faculty are perceived to contribute to internal capacity for educational change in meeting students’ learning and inclusion needs to a greater degree than research focused colleagues. The article also reveals that teaching faculty report significant concerns that include not being provided with adequate training and resources and feeling less included and respected than research-focused faculty within their departments.

Results: RQ1 considers whether an increase has occurred in research expectations for teaching faculty. Results indicate assistant teaching professors report greater research expectations and spending more time on research than their tenured counterparts. The reviewer notes that some mention could be made that this finding might reflect motivation for promotion, in addition to perceived increase in demand.

RQ2 considers how “scholarly activities correlate with professional identity and self-reported influence of TP/PoTs.” Findings indicate that assistant teaching professors engaging in educational research report greater identification as researchers than their senior teaching counterparts and that educational publications increased influence more than teaching and mentoring practices.

RQ3 considers preparedness in terms of training and resources, shedding light on the fact that although teaching faculty are expected to focus on that activity and related research and service, few have received training other than the pedagogical research they conduct. Teaching faculty also report inadequate developmental resources due to lack of departmental awareness that pedagogical research is a form of research that also warrants institutional support. The article concludes with appropriate recommendations for teaching faculty and their institutions to explore strategies to improve training and resources. The article sheds light on important issues that affect faculty, students, and academic departments and suggests that this area of inquiry warrants further research, for which this article provides a foundation.

**Do you want your identity to be public for this peer review?** For information about this choice, including consent withdrawal, please see our Privacy Policy

Reviewer #1: No

Reviewer #2: No

---

## [Author Response · Author response to Decision Letter 1]

29 Jul 2025

PONE-D-25-02548

Reimagining the Role of Teaching-Focused Faculty in Research-Intensive Universities: The Evolution of Scholarly Expectations and Departmental Influence

PLOS ONE

Dear Dr. Paine,

Thank you for submitting your manuscript to PLOS ONE. After careful consideration, we feel that it has merit but does not fully meet PLOS ONE’s publication criteria as it currently stands. Therefore, we invite you to submit a revised version of the manuscript that addresses the points raised during the review process.

Thanks for submitting you article to PLOS ONE. It has been reviewed by two individuals, one who has a review similar to mine and one accept. You need to check the Information for Authors for citations, formatting etc as your manuscript does not follow the guidelines

In addition, it is difficult to understand your analyses as presented.

● Thank you for your feedback. We have responded to initial points regarding analyses below.

Finally, in the discussion and limitations from the findings it seems to me that these teaching professors are not really teaching professors. --they just needed a job. They are more similar to research professors without the support. They are more like those who select to go to traditional teaching colleges. Please discuss the possibility of this in your discussion.

● We somewhat disagree with this point - while true, the PoTs in our sample do have similar training to research faculty, this does not necessarily mean that they did not want their role and/or just needed a job. This is a factor of both what search committees were looking for in candidates (often individuals with similar training as research faculty - as we have seen in prior work by Harlow et. al. 2022) as well as the fact that there currently is not a separate mechanism to train teaching-focused faculty who must possess both disciplinary expertise as well as education-expertise.

● We do agree that if the goal is to bring in more education-trained individuals within this role stakeholders must adjust their hiring expectations. We bring up this point in the Discussion (Recommendations for Hiring).

● A rebuttal letter that responds to each point raised by the academic editor and reviewer(s). You should upload this letter as a separate file labeled 'Response to Reviewers'.

● A marked-up copy of your manuscript that highlights changes made to the original version. You should upload this as a separate file labeled 'Revised Manuscript with Track Changes'.

● An unmarked version of your revised paper without tracked changes. You should upload this as a separate file labeled 'Manuscript'.

We look forward to receiving your revised manuscript.

Kind regards,

Mary Diane Clark, PhD

Academic Editor

PLOS ONE

Journal Requirements:

This work was supported by the National Science Foundation Grant DUE #1821724.

● Per the reviewers request, the funding statement ”This work was supported by the National Science Foundation Grant DUE #1821724, There was no additional external funding received for this study” has been removed from the acknowledgements section and is now only found within the cover letter

This work was supported by the National Science Foundation Grant DUE #1821724.

This work was supported by the National Science Foundation Grant DUE #1821724.

● Per the reviewers request, the funding statement ”This work was supported by the National Science Foundation Grant DUE #1821724, There was no additional external funding received for this study” has been removed from the acknowledgements section and is now only found within the cover letter

4. We note that this data set consists of interview transcripts. Can you please confirm that all participants gave consent for interview transcript to be published?

● All information collected from faculty was done so through the online survey. Quotes were taken from open-response items to this survey. As such consent was provided prior to taking the survey and all survey participants were deidentified prior to data analysis. Information about these open-ended questions can be found throughout the Methods and more text has been added within the manuscript to clarify this form of data collection. Additionally, a statement about deidentification of survey participants has been added to the Methods section.

If they DID provide consent for these transcripts to be published, please also confirm that the transcripts do not contain any potentially identifying information (or let us know if the participants consented to having their personal details published and made publicly available). We consider the following details to be identifying information:

- Names, nicknames, and initials

- Age more specific than round numbers

- GPS coordinates, physical addresses, IP addresses, email addresses

- Information in small sample sizes (e.g. 40 students from X class in X year at X university)

- Specific dates (e.g. visit dates, interview dates)

- ID numbers

Or, if the participants DID NOT provide consent for these transcripts to be published:

- Provide a de-identified version of the data or excerpts of interview responses

- Provide information regarding how these transcripts can be accessed by researchers who meet the criteria for access to confidential data, including:

a) the grounds for restriction

b) the name of the ethics committee, Institutional Review Board, or third-party organization that is imposing sharing restrictions on the data

c) a non-author, institutional point of contact that is able to field data access queries, in the interest of maintaining long-term data accessibility.

d) Any relevant data set names, URLs, DOIs, etc. that an independent researcher would need in order to request your minimal data set.

For further information on sharing data that contains sensitive participant information, please see: https://journals.plos.org/plosone/s/data-availability#loc-human-research-participant-data-and-other-sensitive-data

If there are ethical, legal, or third-party restrictions upon your dataset, you must provide all of the following details (https://journals.plos.org/plosone/s/data-availability#loc-acceptable-data-access-restrictions):

a) A complete description of the dataset

b) The nature of the restrictions upon the data (ethical, legal, or owned by a third party) and the reasoning behind them

c) The full name of the body imposing the restrictions upon your dataset (ethics committee, institution, data access committee, etc)

d) If the data are owned by a third party, confirmation of whether the authors received any special privileges in accessing the data that other researchers would not have

e) Direct, non-author contact information (preferably email) for the body imposing the restrictions upon the data, to which data access requests can be sent

● This does not apply to our dataset

● This does not apply to our dataset

● A heading has been added to the methods to more clearly present this information

Additional Editor Comments:

Important topic as many research trained PhDs are accepting jobs as Teaching faculty most with as little educational backgrounds as those in tenure track research positions.

Reviewer 1 and I have provided many suggestions for improving the manuscript. I have some concerns about how the analyses were conducted as what is provided in the paper is somewhat limited to allow me to evaluate how the statistics were conducted. To avoid this you Data Analyses section needs to be much more detailed to allow the reader to understand your analysis

● We have expanded upon the statistics both in the Methods and Data Analysis sections of this manuscript.

Please read the comments in the attached file.

In addition, there are three sections that I suggest you eliminate. Figure 1 takes a lot of space and could be conveyed easier in text or at most a table. DEI is not the focus on this manuscript, I suggest you take it out. Then Brofenbrenner does not contribute to this manuscript.

● In response to these comments, we have moved Fig 1 to the supplemental materials and references in the Discussion have been removed. In regards to the use of Brofenbrenner, we use this only to cite how his work has influenced the capacity building framework, but can remove this if the editor prefers.

Reviewers' comments:

Reviewer's Responses to Questions

Comments to the Author

1. Is the manuscript technically sound, and do the data support the conclusions?

Reviewer #1: Partly

Reviewer #2: Yes

2. Has the statistical analysis been performed appropriately and rigorously?

Reviewer #1: I Don't Know

Reviewer #2: Yes

3. Have the authors made all data underlying the findings in their manuscript fully available?

Reviewer #1: Yes

Reviewer #2: Yes

4. Is the manuscript presented in an intelligible fashion and written in standard English?

Reviewer #1: Yes

Reviewer #2: Yes

5. Review Comments to the Author

Reviewer #1: The manuscript contains minor grammatical errors that require correction for improved clarity and readability.

● The manuscript has been thoroughly read through to address any grammatical errors.

There are inconsistencies in the formatting and usage of in-text citations that should be addressed.

● Headings of major sections changed to 18pt font and sentence case

● Headings of sub-sections changed to 16pt font … just through methods

● Headings of sub-sections within sub-sections changed to 14pt font

● Document changed to single spaced

● Each paragraph is independent by 5 spaces

The abstract does not include mention of the qualitative or statistical methods used in the study, which may limit readers' understanding of the research design.

● The wording within the abstract has been changed to address this issue

Up to the Methods section, the manuscript lacks any discussion of statistical analysis, which makes the later emergence of a mixed methods approach feel abrupt and disjointed. Earlier sections read as though the study was exclusively qualitative.

● We have added additional references to the survey/quantitative nature of the work prior to the methods to help alleviate this concern within the Abstract and Introduction.

The Discussion section opens with a focus on diversity, equity, and inclusion (DEI), but these themes are not introduced or addressed elsewhere in the manuscript. As a result, this section appears disconnected and underdeveloped.

● This is a great point and we have adjusted the beginning of the Discussion to reflect the need to broadly improve STEM education to support the success of all students.

In the Data Analysis section, the investigators do not address potential statistical relationships between participants’ gender identity and their perceptions of their roles as STEM Teaching Faculty members. Given the near-even gender distribution of participants and the male-dominated natu

---

## [Editor Report · Decision Letter 1]

19 Aug 2025

Dear Dr. Paine,

We look forward to receiving your revised manuscript.

Kind regards,

Mary Diane Clark, PhD

Academic Editor

PLOS ONE

Journal Requirements:

Additional Editor Comments:

Thank you for the major revision. It makes the manuscript clearer. I am not going to send it out for a second review but I do have a few things that I need you to address prior to recommending publication.

In general—you have extremely long sentences. –give it a look a see if some of them can become two or more sentences. One example is below

Very end of page 3 to 4

Comprising roughly 10% of tenure-eligible faculty across the UC system, the promotion criteria for TP/PoT faculty reflect that of the Research Professor (which we define as the traditional academic tenure-track faculty position that is evaluated primarily on the strength of their research program) but with a greater emphasis placed on the value of teaching excellence (Harlow, et. al., 2020, UC 63 University of California, 2020).

Above is a long sentence—if you could take out the definition of traditional academic tenure track faculty it would be easier to parse

Page 4 line 71

Influenced by---add the by

Page 5 83-84

In particular, research on TP/PoT faculty indicates that they are currently doing so 84 within the UC system

Please be more specific about currently doing

Page 5 97-98

it is key that we better understand the role the perspective of TP/PoTs 98 themselves.

Not sure what you mean by the role the perspective

Page 14—why not include the t value with the significance level

What p level did you use given the multiple t tests?

You need to correct the alpha for multiple tests

Same comment for Table 3—did you correct for increased t tests?

I still don’t see the need to take up journal space with the large non-significant effects in Tables 5 and 6

If seems that statements that specific variables are not significant would be enough

I strongly recommend using the tables you now have in the supplemental files witht eh ANOVAs for each and moving these models in to the supplemental data.

I think that readers of this paper will find that more informative

---

## [Author Response · Author response to Decision Letter 2]

29 Sep 2025

Additional Editor Comments:

Thank you for the major revision. It makes the manuscript clearer. I am not going to send it out for a second review but I do have a few things that I need you to address prior to recommending publication.

Point 1: In general—you have extremely long sentences. –give it a look a see if some of them can become two or more sentences. One example is below.

Very end of page 3 to 4

Comprising roughly 10% of tenure-eligible faculty across the UC system, the promotion criteria for TP/PoT faculty reflect that of the Research Professor (which we define as the traditional academic tenure-track faculty position that is evaluated primarily on the strength of their research program) but with a greater emphasis placed on the value of teaching excellence (Harlow, et. al., 2020, UC 63 University of California, 2020).

Above is a long sentence—if you could take out the definition of traditional academic tenure track faculty it would be easier to parse

Response: This specific sentence has been shortened into two separate sentences, and the definition of Research professor has been removed. Additionally, the manuscript has been thoroughly read to locate and address other long sentences, and these have been adjusted for ease of reading.

Point 2: Page 4 line 71

Influenced by---add the by

Page 5 83-84

In particular, research on TP/PoT faculty indicates that they are currently doing so 84 within the UC system

Please be more specific about currently doing

Response: This appears to be a comment on the original manuscript rather than the revised. This edit was made within the revised manuscript and remains revised in this resubmission (theoretical framework paragraph 1). Additionally, the manuscript has been thoroughly read through to check for any additional grammatical errors/missing text such as this.

Point 3: Page 5 97-98

it is key that we better understand the role the perspective of TP/PoTs 98 themselves.

Not sure what you mean by the role the perspective

Response: Thank you for pointing out the lack of clarity within this sentence. As the remainder of the paragraph details this, we have removed this unnecessary and unclear sentence from the manuscript.

Point 4: Page 14—why not include the t value with the significance level

What p level did you use given the multiple t tests?

You need to correct the alpha for multiple tests

Same comment for Table 3—did you correct for increased t tests?

Response: Thank you for this comment. We revised Table 2 to include t-values and degrees of freedom alongside the p-values. We used α = .05 and applied a Holm correction for the six planned comparisons. After correction, expected scholarship (p = .006) and actual service (p < .001) remained significant, while actual scholarship (p = .044) did not.

We revised Table 3 to include t-values and degrees of freedom alongside the p-values. We used an α level of .05 and applied a Holm correction for the nine planned comparisons. After correction, the differences in DBER (p < .001) and K–12 professional development (p = .005) remained significant, while the smaller effects did not.

Point 5: I still don’t see the need to take up journal space with the large non-significant effects in Tables 5 and 6

If seems that statements that specific variables are not significant would be enough

I strongly recommend using the tables you now have in the supplemental files witht eh ANOVAs for each and moving these models in to the supplemental data.

I think that readers of this paper will find that more informative

Response: Thank you for this helpful suggestion. We have moved the full regression models (previously Tables 5 and 6) to the supplemental materials (now S2 and S4 Tables). In the main text, we summarize only the significant predictors and note that all others were non-significant, while highlighting the Type III ANOVA results to confirm the unique contributions.

---

## [Editor Report · Decision Letter 2]

5 Oct 2025

Reimagining the Role of Teaching-Focused Faculty in Research-Intensive Universities: The Evolution of Scholarly Expectations and Departmental Influence

PONE-D-25-02548R2

Dear Dr. Paine,

We’re pleased to inform you that your manuscript has been judged scientifically suitable for publication and will be formally accepted for publication once it meets all outstanding technical requirements.

Kind regards,

Mary Diane Clark, PhD

Academic Editor

PLOS ONE

Additional Editor Comments (optional):

Thank you for these changes. It makes the manuscript easier to follow and to better understand the significant differences.
---

## [Editor Report · Acceptance letter]

PONE-D-25-02548R2

PLOS ONE

Dear Dr. Paine,

I'm pleased to inform you that your manuscript has been deemed suitable for publication in PLOS ONE. Congratulations! Your manuscript is now being handed over to our production team.

Kind regards,

on behalf of

Dr. Mary Diane Clark

Academic Editor

PLOS ONE